# Effect of a Polypropylene Separator with a Thin Electrospun Ceramic/Polymer Coating on the Thermal and Electrochemical Properties of Lithium-Ion Batteries

**DOI:** 10.3390/polym16182627

**Published:** 2024-09-17

**Authors:** Yeongsu Hwang, Minjae Kim

**Affiliations:** Mechanical & Control Engineering, Handong Global University, Pohang 37554, Republic of Korea; ys.hwang@handong.ac.kr

**Keywords:** ceramic coated separator, ionic conductivity, lithium-ion battery safety, polymer binder, wettability, thermal stability, electrospinning

## Abstract

Lithium-ion batteries (LIBs) are well known for their energy efficiency and environmental benefits. However, increasing their energy density compromises their safety. This study introduces a novel ceramic-coated separator to enhance the performance and safety of LIBs. Electrospinning was used to apply a coating consisting of an alumina (Al_2_O_3_) ceramic and polyacrylic acid (PAA) binder to a polypropylene (PP) separator to significantly improve the mechanical properties of the PP separator and, ultimately, the electrochemical properties of the battery cell. Tests with 2032-coin cells showed that the efficiency of cells containing separators coated with 0.5 g PAA/Al_2_O_3_ was approximately 10.2% higher at high current rates (C-rates) compared to cells with the bare PP separator. Open circuit voltage (OCV) tests revealed superior thermal safety, with bare PP separators maintaining stability for 453 s, whereas the cells equipped with PP separators coated with 4 g PAA/Al_2_O_3_ remained stable for 937 s. The elongation increased from 88.3% (bare PP separator) to 129.1% (PP separator coated with 4 g PAA/Al_2_O_3_), and thermal shrinkage decreased from 58.2% to 34.9%. These findings suggest that ceramic/PAA-coated separators significantly contribute to enhancing the thermal safety and capacity retention of high-energy-density LIBs.

## 1. Introduction

Lithium-ion batteries (LIBs) are highly regarded for their efficient energy storage capability and environmental sustainability [1]. Their high energy density and excellent performance are the main reasons for their popularity across various applications [2]. LIBs offer high capacity, a long cycle life, a minimal memory effect, and the ability to operate over a wide range of temperatures, and have therefore found widespread application in portable electronic devices (such as laptops, digital cameras, and mobile phones), as well as in large-scale applications such as electric vehicles (EVs) and energy storage systems (ESS) [3]. However, due to the limitations imposed by their materials, electrolyte stability, electrode thickness, design constraints, and safety concerns, LIBs still have a lower mass-energy density compared to gasoline and diesel [4]. Attempts to address these issues by increasing the electrode density and loading levels to boost the overall energy density were found to introduce manufacturing challenges such as lowering the electrode flexibility, limiting the processing speed, and increasing the likelihood of material breakage. Apart from this, the mobility of lithium-ion is impeded. All of these negatively affect their electrochemical performance [5]. Therefore, the introduction of high-capacity materials that enhance the energy storage without increasing the electrode thickness seems a promising approach to avoid these problems [6]. Researchers aiming to overcome these challenges have been exploring high-nickel cathode materials and silicon anodes, which offer higher specific capacities but require significant changes to the processing techniques. The implementation of these electrodes would make these batteries less viable for immediate industrial application [7]. Consequently, the industry is focusing on more feasible short-term solutions, such as reducing the thickness of the inactive components such as the current collectors and separator, as they strive to increase the volumetric energy density without extensive modifications to current manufacturing processes [8]. Although high-energy-density LIBs provide efficient energy storage solutions, they are also problematic in terms of their safety, particularly as evidenced by fire incidents [9]. Numerous reports indicate that spontaneous combustion and explosions are often the consequence of thermal runaway, a condition that arises when excessive heat within the battery leads to uncontrollable exothermic chemical reactions [10]. These incidents are typically triggered by short circuits and physical damage [11]. For example, mechanical damage to electric vehicles involved in collisions can result in substantial heat release and subsequent thermal runaway, and has been identified as a primary cause of battery fires [12].

At present, the separators that are used in most commercial LIBs are polyolefin-based, typically polyethylene (PE) or polypropylene (PP) [13]. These separators are positioned between the positive and negative electrodes to prevent short circuits caused by direct contact between the electrodes, and they also facilitate the free transmission of lithium ions during battery operation [14]. Although the separator does not directly participate in electrode reactions, its structure and properties are crucial for ensuring battery safety [15]. High electrode densities and loading levels generate more heat during operation, and defects in the separator caused by thermal shrinkage or mechanical damage could lead to internal short circuits, which, in turn, would give rise to the rapid discharge of stored energy [16]. This increased thermal load can elevate the risk of thermal runaway and battery failure, as evidenced by incidents involving high-energy-density batteries [17]. Ensuring the durability of separators under these conditions is critical. Many advanced materials and new manufacturing techniques have been developed to fabricate separators with enhanced heat resistance, mechanical strength, and stability at high temperatures and pressures [18]. For example, novel polymer membranes with excellent thermal stability, such as polyacrylonitrile (PAN), polyimide (PI), polyvinyl alcohol (PVA), polyvinylidene fluoride (PVDF), and polyurethane (PU), have been used to replace traditional polyolefin separators [19]. Lu, Ziheng, et al. highlighted that PI-based separators significantly enhance battery performance by providing higher tensile strength and thermal stability compared to PE separators [20]. Furthermore, Cai et al. showed that crosslinked PVA/citric acid separators offer high ionic conductivity, good electrolyte uptake, and sufficient mechanical strength, which makes them suitable for high-performance LIBs [21]. However, these proposed solutions are associated with high costs and are difficult to immediately apply to current battery designs.

A ceramic composite separator comprises a polyolefin separator matrix and a ceramic layer. These composite separators are formed by applying a coating of inorganic ceramic particles such as alumina (Al_2_O_3_) [22,23,24], silicon dioxide (SiO_2_) [25], titanium dioxide (TiO_2_) [26], and zirconium dioxide (ZrO_2_) [27,28,29] on the surface of the polyolefin separator using a binder, such as PVDF [25], BPO [30], poly(ethylene terephthalate) [23], and PEO [31]. The introduction of the ceramic layer can enhance the absorption of liquid electrolytes and improve the thermal stability because the ceramic particles increase the specific surface area and porosity of the separator to enhance electrolyte uptake and retention [14]. However, the production of PVDF requires the use of organic solvents such as N-methyl pyrrolidone (NMP) and N,N-dimethylformamide, neither of which is environmentally friendly, and they also pose health risks due to their volatile nature [32]. Additionally, because of the poor adhesiveness of PVDF, the inorganic ceramic particles tend to become detached from the separator [33]. In this study, we prepared a composite separator by coating a polyolefin separator with Al_2_O_3_ using a water-based polyacrylate acid (PAA) binder. PAA offers strong adhesion to enhance the stability of the ceramic coating, and its hydrophilic nature improves the electrolyte wettability and ionic conductivity [34]. Moreover, the use of PAA can enhance the mechanical strength and thermal stability of the separator to address the critical challenges associated with conventional polyolefin separators. Various coating methods, such as dip coating, doctor blade coating, atomic layer deposition (ALD) [35], magnetron sputtering [36], and electron beam evaporation, have been reported [29]. To their advantage, dip coating and blade coating are simple and cost-effective processes, and Wu, D. et al. reported that dip-coated Al_2_O_3_ separators exhibited improved thermal resistance [37]. Unfortunately, the inconsistent coating thickness of these separators led to variations in the battery performance and reduced the reliability under high-stress conditions. In comparison, ALD allows for precise film thickness control at the nanometer level, but achieving the necessary hydrophilicity can be challenging [38]. Kim, U et al. noted that enhancing the rate capability of LIBs with ALD-coated separators is difficult due to the substantial number of coating cycles required [39]. Additionally, the prolonged duration of the ALD process poses a significant limitation, because only a few nanometers are deposited per cycle [40]. In contrast, electrospinning is a promising technique for fabricating thin coatings with many pores that increase mechanical strength and improve electrochemical properties [41]. This method employs a high-voltage electric field to fabricate a polymer solution into continuous nanofibers, which are then deposited to create a ceramic/binder coating layer with a much larger specific surface area and porosity compared to dip-coated layers. The high porosity and large specific surface area of electrospun separators provide numerous pathways for ionic transport, thereby significantly enhancing the overall performance and safety of the battery [39]. These advantages make electrospinning a highly effective method for producing advanced LIB separators that meet the demands of modern high-power applications.

In this study, we endeavored to explore the benefits of ceramic/binder-coated separators by integrating an eco-friendly PAA binder with an Al_2_O_3_ ceramic for coating onto a PP separator using the electrospinning technique. This approach enabled us to exploit the significant advantages of electrospinning, namely the formation of a large specific surface area and highly porous structures to enhance the electrolyte uptake and ionic conductivity. Our experiments revealed improvements in the ionic conductivity, mechanical properties, thermal stability, and electrochemical performance of the cells. In thermal shrinking tests, the thermal shrinkage rate of the ceramic composite separators (CCSs) was approximately 23.3% lower than that of the bare separators. Additionally, in coin cell OCV tests, the stability of the separator was extended by 483 s. These advancements address the critical challenges of conventional polyolefin separators, such as poor thermal stability and limited electrolyte wettability, thereby promoting the development of safer and more efficient high-energy-density lithium-ion batteries.

## 2. Materials and Methods

A PP separator with a thickness of 12 µm (NW1240, LUTONG, Cangzhou, China) was used as the coating substrate. Al_2_O_3_ particles (gamma-phase, Sigma-Aldrich, St. Louis, MO, USA) with an average particle size of 20 nm and a polyacrylic acid (PAA) binder (average Mv~450,000, Sigma-Aldrich) were used to prepare the coating slurry. The slurry was prepared by mixing 0.5 g of Al_2_O_3_ with 5 g of ethanol using a Thinky mixer (ARM-310, THINKY, Tokyo, Japan). The mixture was subjected to a speed of 2000 rpm for 30 min to ensure that the Al_2_O_3_ particles were uniformly dispersed. Following the initial mixing, different batches were prepared by adding different amounts of PAA (8 wt.% in deionized (DI) water) to the Al_2_O_3_ suspension in ethanol (0.5, 1, 2, 3, and 4 g of PAA, respectively). The resulting slurry was coated onto one side of the porous PP separator by an electrospinning machine (ESR100D, NanoNC, Seoul, South Korea) to form a randomly distributed coating layer of ceramic/PAA binder with a thickness of approximately 5 μm. The mixture was electrospun using a 20 mL syringe with a 21gauge needle with an inner diameter of 0.5 mm. The distance between the needle and the bare PP substrate was maintained at 15 cm, and a constant flow rate 1.8 mL h^−1^ was chosen at a constant voltage 12 kV. Additionally, the speed of the rotating drum collector was set at 500 rpm. The collected composite was dried in an air-circulated oven at 40 ℃ overnight to evaporate the solvent and ensure proper adhesion of the Al_2_O_3_ particles to the PP substrate. All experiments were conducted under standard laboratory environmental conditions.

### 2.1. Fabrication of Electrode and Cell Assembly

The anode electrodes were prepared using a slurry containing 80 wt.% artificial graphite (SCMG-BH, Showa Denko, Tokyo, Japan), 10 wt.% Super-P (DW-05, IMERYS Graphite&Carbon, Paris, France), 6 wt.% carboxymethyl cellulose (CMC, Daicel 2200, Tokyo, Japan), and 4 wt.% styrene–butadiene rubber (SBR, BM-451B, ZEON, Tokyo, Japan), with DI water as the solvent. The slurry was cast onto copper foil (10 µm thickness) using the doctor blade technique. The coated electrodes were dried in a drying oven at 40 ℃ and then roll-pressed using a gap-control-type roll pressing machine (Rohtec, Gwangju, South Korea) to ensure precise control over the thickness, density, and loading amount. The resulting anode electrodes had a loading level of 3 mg cm^−2^, a density of 1.3 g cm^−3^, and a thickness of 33 µm.

The cathode electrodes were prepared by mixing 80 wt.% NCM 811 (L&F Co., Ltd., Daegu, South Korea), 10 wt.% Super-P, and 10 wt.% polyvinylidene fluoride (PVDF) dissolved in N-methyl-2-pyrrolidone (NMP). The slurry was cast onto aluminum foil. The coated electrodes were dried in a drying oven at 60 ℃ and roll-pressed. The resulting cathode electrodes had a loading level of 4.33 mg cm^−2^, a density of 3.5 g cm^−3^, and a thickness of 30 µm.

For the electrochemical performance tests, 2032-type coin half-cells were assembled using the prepared anode and lithium metal. The electrolyte used was a commercial solution containing 1 M LiPF_6_ in a 1:1 volume ratio of ethylene carbonate (EC) and diethyl carbonate (DEC), with an additional 5% fluoroethylene carbonate (FEC). The cells were assembled in an argon-filled glove box.

The high-temperature OCV tests were conducted with full cells that were assembled using the prepared graphite anode and the NCM 811 cathode by maintaining an N/P ratio of 1.1. The same electrolyte solution was used, and the assembly was conducted in the argon-filled glove box.

### 2.2. Characterization of the Separators

The surface morphology and thickness of the ceramic-coated separators were analyzed using field emission scanning electron microscopy (FE-SEM, SU8230, Hitachi, Tokyo, Japan).

To examine the mechanical properties, the mechanical strength was evaluated using a multi-axis texture analyzer (TXA texture analyzer, YEONJIN S-Tech Corp., Yongin, Republic of Korea) at a strain rate of 0.5 mm s^−1^.

Square separators with a length of 3 cm, sandwiched between two glass plates, were subjected to 140 °C for 0.5 h for the heat treatment. The thermal shrinkage of the separator was calculated according to the following equation:(1)$$Shrinkage\;\left(\%\right)=\frac{S_{0}-S}{S_{0}}\times 100\%$$
where *S*_0_ and *S* are the areas of the membranes before and after the heat-treatment test, respectively.

Static contact angle measurements of the separators, as one of the indicators for the wettability of the separator with the electrolyte, were recorded using a digital camera (700D, Canon, Tokyo, Japan).

The ionic conductivity of the separators was investigated by fabricating blocking-type cells by sandwiching the liquid electrolyte-soaked separators between two stainless steel electrodes, whereupon the impedance data of the cells were measured by a potentiostat (SP-150e, BioLogic, Seyssinet-Pariset, France) with a frequency range from 1 Hz to 100 kHz. Ionic conductivity can be calculated with the following equation:(2)$$\sigma =\frac{L}{R_{b}\times A}$$
where σ is the ionic conductivity, $R_{b}$ is the bulk resistance, *L* is the thickness of the separator and *A* is the area of the stainless-steel electrode.

The variation in the alternating current (AC) impedance of the cells with cycling was investigated by measuring the AC impedance data of the graphite cells prepared with the PP separator and CCS using the potentiostat (SP-150e, BioLogic, Seyssinet-Pariset, France) in the frequency range from 0.1 Hz to 100 kHz.

The safety performance of cells assembled with the bare PP separator and ceramic-coated separators, respectively, was assessed by carrying out open circuit voltage (OCV) measurements. Coin cells were assembled by sandwiching the separators between NCM 811 cathodes and graphite anodes. The cells were charged to 4.3 V at room temperature and then placed in a drying oven at 150 ℃, after which the OCV of the cells was simultaneously monitored as a function of time using the potentiostat (SP-150e, BioLogic, Seyssinet-Pariset, France).

### 2.3. Electrochemical Tests

Electrochemical impedance spectroscopy (EIS) was performed using the potentiostat with a frequency response analyzer. The measurement was conducted over a frequency range of 100 kHz to 0.1 Hz with an AC amplitude of 10 mV.

All half-cells were subjected to a formation procedure of three charge and discharge cycles at a C-rate of 0.1 C to sufficiently form the solid electrolyte interphase (SEI). Discharging was conducted using a constant current/constant voltage (CCCV) operation with a cutoff current of 0.025 C, and the cells were charged in constant current (CC) mode within a voltage window of 0.01–3 V.

After formation, the half-cells underwent a cycling test of 150 cycles. The cells were discharged with a CCCV procedure, following a discharge current of 0.5 C and 1 C until the cutoff voltage of 0.01 V and a cutoff current of 0.025 C in the CV phase. Charging proceeded in CC mode at 0.5 C and 1 C charge current until the cutoff voltage of 3 V was reached. The galvanostatic charging/discharging of half-cells was also conducted in the potential window of 0.01–3 V vs. Li/Li^+^ with different current densities. Following the formation step, the cells were cycled at different C-rates (0.2 C, 0.5 C, 1 C, 2 C, 3 C, 5 C) to evaluate their rate capability. Each rate was applied for five cycles to ensure stable and repeatable measurements. All electrochemical measurements were carried out on a Neware battery test system (CT-4008T-5V50mA-164, Neware, Shenzhen, China) at constant room temperature.

## 3. Results

As shown in Figure 1, the electrospinning process is simple and useful for fabricating the ceramic-coated layer on the PP separator. This process involves coating the polypropylene (PP) separators with a solution containing Al_2_O_3_ ceramic particles and a polyacrylic acid (PAA) binder. The electrospinning method can produce a thin ceramic layer with randomly dispersed particles while increasing the porosity of the separator, which helps to maintain the required thermal and mechanical properties to maximize the energy density of lithium-ion batteries (LIBs). Additionally, the electrospun coating enhances the wettability and ionic conductivity, both of which are crucial for improving the overall electrochemical performance of the batteries [32]. The Assessment of the morphological and compositional characteristics of the ceramic-coated separators (CCS) is essential to assess their effectiveness.

The scanning electron microscopy (SEM) images in Figure 2 show that the Al_2_O_3_ particles are randomly dispersed on the surface of the ceramic-coated separators. This random distribution with high porosity is crucial for maintaining the structural integrity and performance of the separators. Randomly distributed Al_2_O_3_ particles serve to enhance the absorption and ionic conductivity of the electrolyte to improve the electrochemical performance of the battery. The SEM images also demonstrate the morphological changes in the coating surface as the PAA binder content increases. At a lower PAA content, the binder appears to have zero-dimensional dot-like formations, whereas at higher concentrations, it transitions into one-dimensional continuous line structures. This transformation enhances the coating uniformity and strengthens the structural integrity of the coating layer.

Specifically, the 0.5 g PAA-coated separator (0.5g-PAA/Al_2_O_3_, Figure 2b) has a smooth, homogeneous appearance, with randomly dispersed ceramic particles, which maintains the porosity [42]. This increases the absorption and retention of the electrolyte and facilitates the movement of lithium ions, thereby improving the battery performance. As the PAA content increases to 1 g (1g-PAA/Al_2_O_3_, Figure 2c), the binder transitions from dot-like formations to continuous line structures. With further increases in the PAA content, the polymer binder forms fibers that strengthen the adhesion between the separator and the ceramic particles to enhance the mechanical strength. Apart from the additional mechanical strength, the thermal stability is enhanced, which improves the durability of the battery.

Starting from the 2g-PAA/Al_2_O_3_ separator (Figure 2d), the fiber structure becomes more pronounced, with the binder increasingly forming more fiber structures and the ceramic particles being distributed in larger clusters. In the 3g-PAA/Al_2_O_3_ separator, even larger clusters of ceramic particles are formed, and the ceramics are distributed between the fibers. The 4g-PAA/Al_2_O_3_ separator has the largest clusters of ceramic particles, although the coating uniformity is significantly reduced. Nevertheless, the highest PAA content provides maximum mechanical strength and thermal stability. Keeping in mind that a high PAA content may hinder ionic conductivity, it is essential to find the optimal PAA content for the best performance.

The open circuit voltage (OCV) tests were conducted on full cells with the NCM/graphite electrode employing a bare PP separator and CCSs while the cells were exposed to heat at 150 ℃. Prior to this, the cells were fully charged to 4.3 V at room temperature. The tests demonstrated that the cells with the CCSs maintained stable voltages significantly longer than those with the bare PP separators. Specifically, the OCV of cells with the bare separator dropped to 0 V after only 453 s, whereas that of cells with the 0.5g-PAA/Al_2_O_3_ separator decreased to 0 V after 537 s. As the PAA content increased, the time to reach 0 V became longer: 1g-PAA/Al_2_O_3_ at 569 s < 2g-PAA/Al_2_O_3_ at 737 s < 3g-PAA/Al_2_O_3_ at 871 s < 4g-PAA/Al_2_O_3_ at 937 s (Figure 3a). These results highlight the superior thermal stability and safety of the ceramic-coated separators that were fabricated using the electrospinning process by preventing internal short circuits caused by thermal shrinkage within the battery.

The tensile strength was measured to evaluate the mechanical properties of the ceramic/binder-coated separators. As shown in Figure 3b, the extent to which the separators could undergo elongation increased as the PAA binder concentration increased. Specifically, the tensile strength and elongation values of the separators were measured as follows: for 0.5g-PAA CCS, the tensile strength was 261.8 MPa, and the elongation was 136.8%; for 1g-PAA CCS, the tensile strength was 263.4 MPa, and the elongation was 123.9%; for 2g-PAA CCS, the tensile strength was 269.0 MPa, and the elongation was 128.4%; for 3g-PAA CCS, the tensile strength was 268.4 MPa, and the elongation was 128.6%; and for 4g-PAA CCS, the tensile strength was 268.1 MPa, and elongation was 129.1%. This improvement indicates that, as the PAA content increases, the flexibility of the separator improves, thereby providing clear evidence of enhanced mechanical stability. Additionally, the higher PAA polymer content strengthened the adhesiveness of the ceramic coating layer. These results confirm that the mechanical properties of the ceramic-coated separators have been improved.

The main roles of separators are to prevent electrical short circuits by preventing the anode and cathode from contacting each other and to conduct lithium ions when soaked in the electrolyte [14]. Therefore, separators have to maintain their stability even in battery-related accidents. However, conventional PE and PP separators, which are manufactured through multiple stretching processes, can easily undergo dimensional changes in high-temperature environments. This can lead to thermal runaway, which could cause fires or explosions in cells equipped with these separators [13]. To measure the rate at which the separators underwent thermal shrinkage at high temperatures, we exposed the separators to a high temperature for half an hour. Photographs of these separators are shown in Figure 3c. During the high-temperature test, the separators (3 cm × 3 cm) were sandwiched between two glass plates. This setup ensured that the asymmetric curvature of the separators coated on one side was effectively suppressed during the high-temperature storage test. The thermal shrinkage of 0.5g-PAA/Al_2_O_3_, 1g-PAA/Al_2_O_3_, 2g-PAA/Al_2_O_3,_ 3g-PAA/Al_2_O_3_, and 4g-PAA/Al_2_O_3_ was quantified as approximately 54.1%, 51.4%, 41.5%, 38.9%, and 34.9%, respectively, in comparison to the thermal shrinkage of the bare PP separator of 58.2%. The photographs clearly show that the bare PP separator experiences a high degree of dimensional change after exposure to high temperatures. Furthermore, the shrinkage of the ceramic/binder-coated separators clearly decreased with increasing concentrations of the PAA binder, with the 4g-PAA/Al_2_O_3_ membrane having undergone the lowest degree of dimensional change. These results confirmed that the thermal stability of the separator could be effectively enhanced by introducing ceramic/binder composite coating.

The effect of the separator coating on the electrochemical performance of the cell was analyzed by assessing the capacity retention during cycling using 2032 coin-type half-cells based on graphite and Li metal. As shown in Figure 4a, after precycling, the unit cells based on bare PP separators delivered 353.95 mAh g^−1^, whereas the cells with separators containing xg-PAA/Al_2_O_3_ (x = 0.5, 1, 2, 3, and 4) delivered 361.51, 355.08, 365.32, 359.35, and 360.81 mAh g^−1^, respectively. This indicates that the ceramic-coated separators with the Al_2_O_3_/PAA coating had almost identical charge–discharge curves to the bare PP separator.

The capacity retention with cycling was tested by operating the unit cells at the 0.5 C and 1 C rates for both the charging (CC/CV mode) and discharging (CC mode) processes at room temperature. As shown in Figure 4b,c, the respective unit cells maintained their initial discharge capacity up to 150 cycles. These results indicate that the surface coatings did not significantly impede the electrochemical performance of the cells, and the cells with the coated separators exhibited excellent cycling stability and capacity retention.

The Nyquist plot in Figure 5 shows the impedance characteristics of cells containing separators with the xg-PAA/Al_2_O_3_ (x = 0, 0.5, 1, 2, 3, and 4) coatings. In the high-frequency region, the semicircle represents the charge transfer resistance (*R*_ct_) at the electrode/electrolyte interface [14]. The semicircle of 0.5g-PAA/Al_2_O_3_ is the smallest, which represents the lowest charge transfer resistance and suggests superior electrochemical performance compared to the bare PP separator [19]. As the PAA content increases, the diameter of the semicircle gradually increases, with the 1.0g-PAA/Al_2_O_3_ CCS having a slightly larger semicircle, but still significantly smaller than that of the bare PP separator, hence indicating lower resistance. This trend continues for the 2g-PAA/Al_2_O_3_, 3g-PAA/Al_2_O_3_, and 4g-PAA/Al_2_O_3_CCSs, with increasing semicircle sizes but performance superior to that of the bare PP separator. In the low-frequency region, the linear portion corresponds to the Warburg impedance, which is associated with lithium-ion diffusion within the electrode material [11]. The slopes of these lines of all the CCSs are steeper than those of the bare PP separator, suggesting enhanced ionic conductivity and improved lithium-ion diffusion in the cells with CCSs. These observations indicate that incorporating a ceramic/PAA composite layer significantly enhances the electrochemical properties of lithium-ion cells. CCSs with lower PAA content markedly lower the charge transfer resistance and improve the lithium-ion diffusion, whereas those with higher PAA content slightly increase the resistance but still deliver superior performance compared to the bare PP separators. The 0.5g-PAA/Al_2_O_3_ CCS exhibits the best overall performance. This suggests that ceramic/PAA composite separators can increase the efficiency and stability of LIBs, which makes these separators a promising solution for high-performance energy storage applications. Additionally, to further confirm the accuracy of the impedance analysis, the EIS spectra have been fitted by exploiting the equivalent circuit show in Appendix A. The fitting was performed using a equivalent circuit model to represent the impedance characteristics, and the fitted data closely matched the experimental results. This agreement confirms the accuracy of the proposed model in capturing the electrochemical behavior of the cells, further supporting the observed trends in charge transfer resistance and lithium-ion diffusion.

The rate capability in Figure 6a of half-cells with the bare PP and ceramic/binder separators were also evaluated. The cells were charged to 3 V and discharged to 0.01 V at current rates of 0.2 C, 0.5 C, 1 C, 2 C, 3 C, and 5 C. All cells had an initial capacity of approximately 350 mAh g^−1^ at 0.2 C. At 0.5 C, the capacity of all the cells was slightly lower, with the cell with the 0.5g-PAA/Al_2_O_3_ CCS maintaining the highest capacity. At 1 C, the capacity of all the cells further decreased, but the cell with the 0.5g-PAA/Al_2_O_3_ CCS still exhibited the best performance. At 2 C and 3 C, the capacity continued to decline, and at 5 C, the capacity of all the cells decreased sharply. Finally, upon returning the current rate to 0.2 C, all the cells recovered to near their initial capacity. The cell with the 0.5g-PAA/Al_2_O_3_ CCS demonstrated relatively higher capacity retention even at high current densities. Notably, the cells with the 0.5g-PAA/Al_2_O_3_ and 1.0g-PAA/Al_2_O_3_ CCSs delivered superior rate performance compared to the cells with the other CCSs across the range of current densities. However, the rate capability tends to increase as the PAA binder content decreases. This could be due to the higher electrolyte uptake and retention capabilities and larger pores, which would enhance the ionic conductivity and increase the charge and discharge capacities at high current rates [43].

The Nyquist plot in Figure 6b demonstrates the ionic conductivity of cells equipped with the different separators xg-PAA/Al_2_O_3_ (x = 0, 0.5, 1, 2, 3, and 4). Here, the high-frequency intercept on the real axis (Re(Z)) represents the bulk resistance of the electrolyte and separator. As summarized in Table 1, the bare PP separator had the highest resistance at 3.87 Ω, resulting in the lowest ionic conductivity of 0.321 mS/cm. In contrast, the CCSs with the PAA binder had significantly lower resistance levels and hence higher ionic conductivities. The 0.5g-PAA/Al_2_O_3_ CCS gave rise to resistance of 1.71 Ω and ionic conductivity of 0.728 mS/cm, whereas the 1g-PAA/Al_2_O_3_ CCS had the highest ionic conductivity of 0.852 mS/cm with resistance of 1.46 Ω. The cells with xg-PAA/Al_2_O_3_ (x = 2, 3, and 4) exhibited resistances of 1.81, 1.89, and 2.08 Ω, with corresponding ionic conductivities of 0.687, 0.658, and 0.598 mS/cm, respectively. These results indicate that increasing the PAA binder content of the CCSs enhances the ionic conductivity up to a point, after which further increases lower the conductivity slightly. The enhancement of the ionic conductivity by optimizing the PAA content is attributed to the improved wettability and electrolyte uptake, which facilitate more efficient ionic transport. These findings suggest that PAA/Al_2_O_3_ CCSs can significantly improve the efficiency and performance of LIBs to render them suitable for high-performance energy storage applications.

Interaction between the separator and electrolyte is essential for optimal battery manufacturing and performance. High wettability ensures that the electrolyte filling process during assembly is faster and more efficient [14]. Additionally, it improves the ability of the separator to retain the electrolyte and enhances ion transport between the electrodes during battery operation. The wettability of the liquid electrolyte on the surfaces of bare PP and ceramic-coated separators with varying PAA binder contents was evaluated by conducting contact angle measurements, as shown in Figure 7. The contact angle of the bare PP separator (42.8°) indicated poor wettability. The addition of increasing amounts of the PAA binder had the effect of significantly decreasing the contact angle: 20.9° for 0.5g-PAA/Al_2_O_3_, 7.4° for 1g-PAA/Al_2_O_3_, and 0° for 2g-, 3g-, and 4g-PAA/Al_2_O_3_, which demonstrated complete wettability. These results indicate that increasing the PAA content of the ceramic-coated separators markedly enhances the electrolyte wettability.

## 4. Conclusions

In conclusion, we successfully prepared a new type of ceramic-coated separator for lithium-ion batteries by depositing an Al_2_O_3_ ceramic layer with a polyacrylic acid (PAA) binder onto a polypropylene (PP) separator using the electrospinning technique. Our results demonstrate that the ceramic-coated separators exhibit superior thermal stability, improved wettability, and high ionic conductivity compared to the bare PP separator. Electrochemical tests with 2032-coin cells revealed that the rate capability of the cells containing the ceramic-coated separators was significantly enhanced. However, as the PAA content of the coating increased, the cell performance gradually decreased at high C-rates. Based on the ionic conductivity and EIS results, higher PAA concentrations were found to block the pores of the separator, thereby hindering ion conduction and increasing the interfacial resistance with the electrolyte. OCV tests indicated that these separators maintained stable voltages significantly longer during heat exposure, with the 4g-PAA/Al_2_O_3_ CCS lasting twice as long as the bare PP separator. Additionally, tensile strength and thermal shrinkage tests confirmed improved mechanical properties and dimensional stability. Contrary to the electrochemical evaluations, higher PAA content further improved the mechanical properties and thermal stability. These findings suggest that PAA-enhanced ceramic-coated separators prepared via electrospinning offer a promising solution for developing safer and more efficient high-energy-density lithium-ion batteries, thereby addressing critical challenges associated with traditional polyolefin separators.

## Figures and Tables

**Figure 1 polymers-16-02627-f001:**
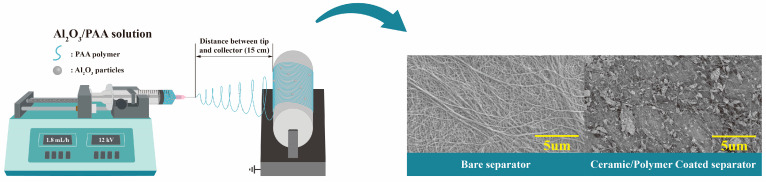
Schematic illustration of the fabrication of the ceramic/polymer coated separator with a simple electrospinning process.

**Figure 2 polymers-16-02627-f002:**
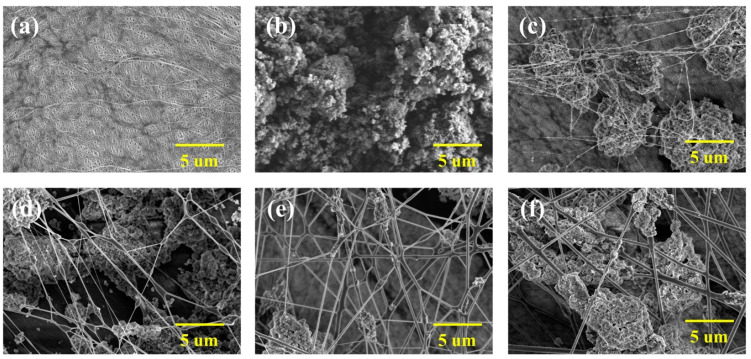
FE-SEM images of (**a**) bare PP separator, (**b**) 0.5g-PAA/Al_2_O_3_, (**c**) 1g-PAA/Al_2_O_3_, (**d**) 2g-PAA/Al_2_O_3_, (**e**) 3g-PAA/Al_2_O_3_, and (**f**) 4g-PAA/Al_2_O_3_ ceramic coated separators.

**Figure 3 polymers-16-02627-f003:**
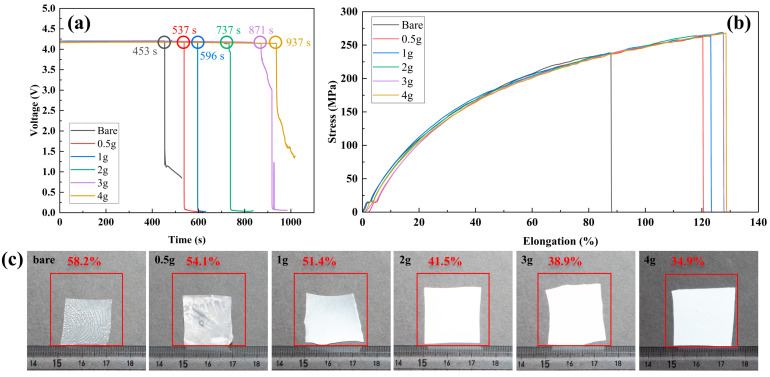
(**a**) Variation in the OCV of the NCM 811/graphite cells employing the bare PP separator and ceramic/binder-coated separators measured at 150 °C. (**b**) Tensile strength: stress-elongation curve for bare PP separator and CCSs with different PAA contents. (**c**) Thermal shrinkage: photographs of the bare PP separator and PAA/Al_2_O_3_-coated separators after heat treatment at 150 °C for 0.5 h.

**Figure 4 polymers-16-02627-f004:**
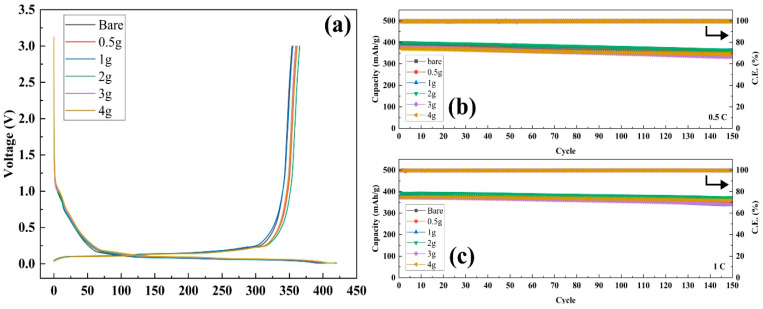
(**a**) First charge/discharge profiles of graphite half-cells prepared with separators with different compositions. Cycling performance of the coin cells with the PP separator and ceramic/binder separators at (**b**) 0.5 C and (**c**) 1 C. The arrow represents Coulomb efficiency.

**Figure 5 polymers-16-02627-f005:**
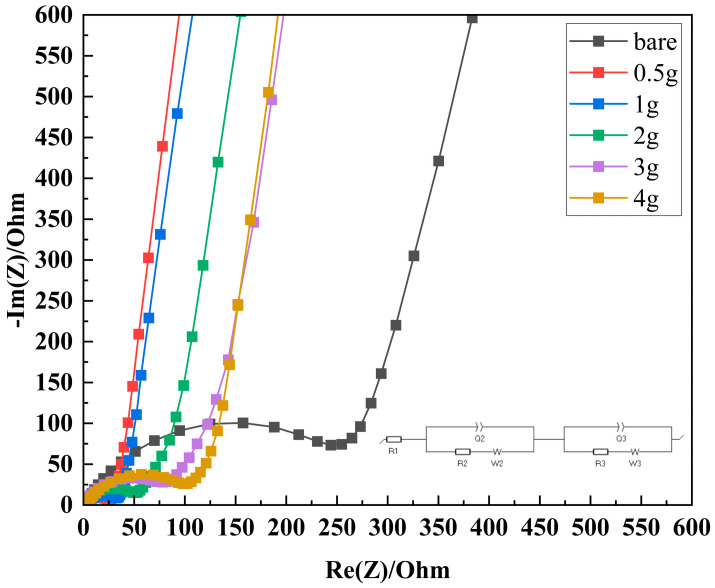
AC impedance measurements of lithium-ion cells assembled with different separators.

**Figure 6 polymers-16-02627-f006:**
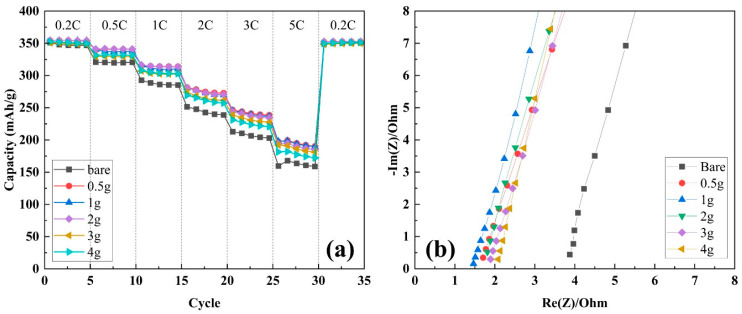
(**a**) Rate performance of cells with separators coated with xg-PAA/Al_2_O_3_ (x = 0, 0.5, 1, 2, 3, and 4) and (**b**) ionic conductivities of cells with separators coated with xg-PAA/Al_2_O_3_ (x = 0, 0.5, 1, 2, 3, and 4) and soaked in liquid electrolyte.

**Figure 7 polymers-16-02627-f007:**
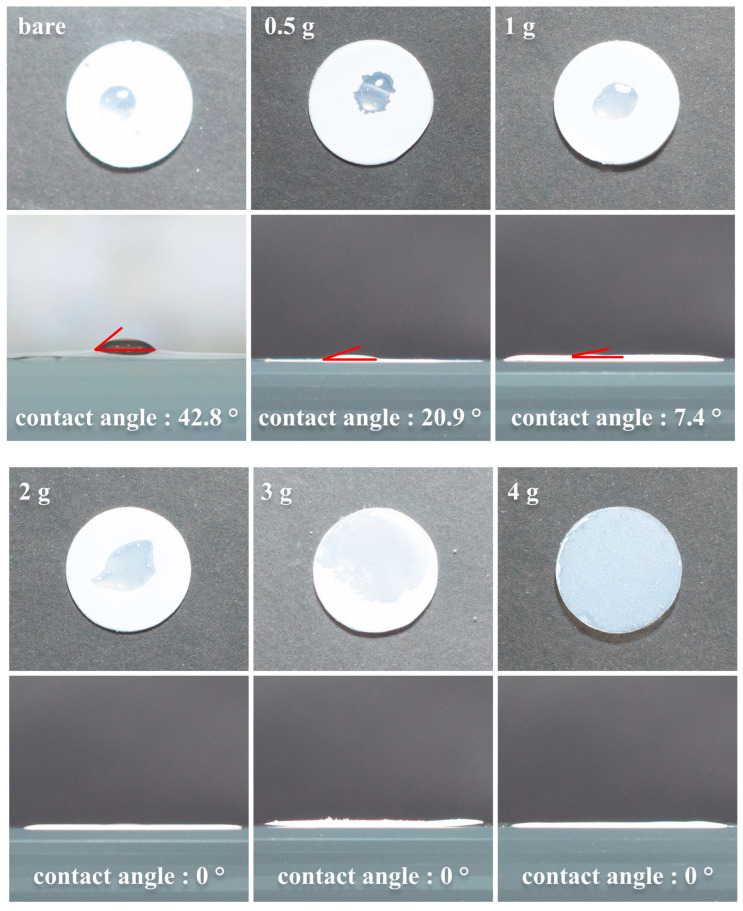
Photographs to demonstrate the wettability and contact angle with the liquid electrolyte (EC:DEC = 1:1 *v/v* containing 1 M LiPF_6_ + 5% FEC) of separators coated with xg-PAA/Al_2_O_3_ (x = 0, 0.5, 1, 2, 3, and 4).

**Table 1 polymers-16-02627-t001:** Ionic conductivity of cells containing the bare separator and those coated with xg-PAA/Al_2_O_3_ (x = 0.5, 1, 2, 3, and 4).

	Bare PP	0.5 g CCS	1 g CCS	2 g CCS	3 g CCS	4 g CCS
R_bulk_ (Ω)	3.87	1.71	1.46	1.81	1.89	2.08
Ionic conductivity (mS cm^−1^)	0.32	0.73	0.85	0.69	0.66	0.60

## Data Availability

Data are available from the corresponding author upon reasonable request.

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
