# Peer review of "Effect of a Polypropylene Separator with a Thin Electrospun Ceramic/Polymer Coating on the Thermal and Electrochemical Properties of Lithium-Ion Batteries"

_polymers, 2024, doi:10.3390/polym16182627_

Round 1

Reviewer 1 Report

Comments and Suggestions for Authors

In the work, authors report a new type of ceramic-coated separator for LIBs by depositing an Al2O3 ceramic layer with a PPA binder on to a PP separator. The as-obtained hybrid separators display competitive electrochemical Li-storage behaviors. All in all, the work is interesting. However, several modifications are still needed as follows for better readership.

1.      The stable electrochemical window of the resulted separators should be well checked, which is of great significance to its applications in LIBs with different cathodes.

2.      The high-/low-temperature properties of LIBs with the optimized separators should be provided in the revised version, such as 50 and -20 0C.

3.      More discussions about the role of Al2O3 in the enhanced lithium-storage behaviors should be well made.

Comments on the Quality of English Language

necessary enhancement is needed

Reviewer 2 Report

Comments and Suggestions for Authors

In this work, a new type of ceramic-coated separator was prepared to enhance the thermal and electrochemical properties of lithium-ion batteries, where the AlO particles with polyacrylic acid (PAA) binders were deposited onto the polypropylene (PP) separator by using the electrospinning technique. The improvements were significant compared with the bare PP separator. It is an interesting research work, which can catch readers’ attention. However, the introduction part is too long, and the result part is very short, lacking critical and comparative evaluation of literatures, as well as discussion and analysis of the experimental results.

 Suggestions: 

(1) Minor editing of English language is required. There are grammatical mistakes and long sentences in the manuscript that are incomprehensible. For example, the ‘mechanical’ in line 12 should be changed to a noun rather than an adjective. The first ‘C-rates’ in line 15 should not be abbreviated. Please modify them and also infer other mistakes from this fact. 

(2) In abstract, the C-rates efficiency of cells with 0.5 g PAA/AlO separator was approximately 10.2% higher than the cells with bare PP separator. Then, the stability, elongation and thermal shrinkage of the 4 g PAA/AlO separator rather than the 0.5 g PAA/AlO separator were used for the comparison with the bare PP separator. Although the improvements were significant, they did not appear in the same separator. Please make corresponding choices and modify the properties to the optimal separator.

I also wonder which separator you recommend mostly in the practical production and application, and please add the reason for the optimal separator in the conclusion part. 

(3) In line 80-82, please illustrate the reason for the high cost and the difficulty for the immediate application. If possible, please quantify them. 

(4) In introduction, can you please provide a comprehensive introduction highlighting the status of the research work in ceramic composite separators, including not only the ceramic particles but also other binders besides PVDF. The corresponding literatures are also too few.

Although the introduction part is long, the introduction of ceramic composite separators is incomplete. In lines 100-102, many coating methods were introduced, but only a few of them were discussed. Please modify them. If possible, please also introduce the challenges in preparing ceramic composite separators through electrospinning and highlight the advantages of electrospinning in this manuscript compared with other electrospinning.

 (5) The ‘surface area’ in the manuscript should be ‘specific surface area’. Because the surface area is related to the area of separators.

 (6) In lines 127-129, ‘the thermal shrinkage rate of the ceramic composite separators (CCSs) was approximately 23.3% higher than that of the bare separators’. The higher should be changed to the lower. Please carefully check the contents of the manuscript.

 (7) In line 130, please change the 500 s to a specific number.

 (8) In Figure 1, please add the scale for the SEM image.

 (9) In lines 248-250, why evenly distributed AlO particles can serve to enhance the absorption and ionic conductivity of the electrolyte to improve the electrochemical performance of the battery? Please explain this. By the way, the AlO particles seem not very uniform in Figures 1 and 2.

Why you choose the AlO particles rather than other ceramic particles? And, please the add the advantages of AlO particles for the thermal and electrochemical properties of lithium-ion batteries. If the reason is that the ceramic particles can increase the surface area and porosity of the separator to enhance electrolyte uptake and retention, the use of AlO particles in this manuscript is not very convincing. It seems that all the high surface area and porosity particles could reach such an effect. Because there seems to be no discussion about the existence of AlO particles on the performance of ceramic composite separators.

 (10) The shrinkage of 4g-PAA/Al2O3 in Figure 3c is too fuzzy. Please modify it.

 (11) There is no Figure 6a in lines 376-378. Please modify them and also infer other mistakes from this fact.

 (12) In lines 387-388, the rate capability tends to decrease as the PAA binder content increases. Then, in lines 388-391, it is confusing that this reason seems unrelated to lines 387-388. Please check this.

 (13) In references, I can’t find the literature [26] by entering the title. Furthermore, the DOI number also correspond to another literature (‘Low overpotential and high current CO2 reduction with surface reconstructed Cu foam electrodes’). On the other hand, the ‘Nano Energy, 2016, 27, 27-34’ is also wrong (‘Layered P2-Na2/3[Ni1/3Mn2/3]O2 as high-voltage cathode for sodium-ion batteries: The capacity decay mechanism and Al2O3 surface modification’). This mistake was found by spot check. Therefore, I doubt that there may be other mistakes in the reference part. Please modify them carefully and also infer other mistakes from this fact.

 (14) What are the novel points in this manuscript?

Comments on the Quality of English Language

The quality of language needs to be improved. 

Reviewer 3 Report

Comments and Suggestions for Authors
  • Dear author, please pay attention to the comments
  • Lack of References: The introduction provides some references, but it lacks a comprehensive review of previous research. Citing a broader range of sources would provide a more solid foundation for the study and better justify the research approach.
  • Typographical and Grammatical Errors: The text appears well-edited overall, but proofread is recommended to catch minor typographical or grammatical.
  • Equations and Formulas: Ensure all equations are correctly formatted and consistent throughout the manuscript.
  • Lack of Control Experiments: The details about preparation and testing of ceramic/polymer-coated separators in the manuscript is not enough.
  • Reproducibility: The methods section provides a detailed account of the experimental procedures. However, to ensure reproducibility, the manuscript should include more information about environmental conditions during experiments or variations in material properties.
  • Detailed Methodology: The methods section should include more detailed descriptions of the experimental setups, including calibration procedures for equipment and specific environmental conditions (at least temperature).
  • Data Interpretation: The manuscript presents various results, including thermal stability and mechanical properties. However, deep analysis of the data should be done. Results of TGA for the samples should be added to the manuscript.
  • The conclusions should be supported by the data in details.
Comments on the Quality of English Language

Extensive editing of English language required.
